# Detection of *Pneumocystis jirovecii* in Hospitalized Children Less Than 3 Years of Age

**DOI:** 10.3390/jof7070546

**Published:** 2021-07-09

**Authors:** Estelle Menu, Jean-Sélim Driouich, Léa Luciani, Aurélie Morand, Stéphane Ranque, Coralie L’Ollivier

**Affiliations:** 1Laboratoire de Parasitologie-Mycologie, IHU Méditerranée Infection, 13385 Marseille, France; jean-selim.driouich@ap-hm.fr (J.-S.D.); Stephane.ranque@ap-hm.fr (S.R.); Coralie.lollivier@ap-hm.fr (C.L.); 2Institut de Recherche pour le Développement, Aix Marseille Université, Assistance Publique-Hôpitaux de Marseille, Service de Santé des Armées, VITROME: Vecteurs-Infections Tropicales et Méditerranéennes, 13385 Marseille, France; 3Unité des Virus Emergents (UVE), Aix Marseille Université, IRD 190, INSERM 1207, IHU-Méditerranée Infection, 13385 Marseille, France; lea.luciani@ap-hm.fr; 4Institut de Recherche pour le Développement IRD, Aix-Marseille Université, Assistance Publique-Hôpitaux de Marseille, AP-HM, MEPHI: Microbes, Evolution, Phylogénie et Infection, IHU-Méditerranée-Infection, 13385 Marseille, France; aurelie.morand@ap-hm.fr

**Keywords:** *Pneumocystis jirovecii*, infants, co-infection, risk factor

## Abstract

Few data are available in the literature regarding *Pneumocystis jirovecii* infection in children under 3 years old. This retrospective cohort study aimed to describe medically relevant information among them. All children under 3 years old treated in the same medical units from April 2014 to August 2020 and in whom a *P. jirovecii* evaluation was undertaken were enrolled in the study. A positive case was defined as a child presenting at least one positive PCR for *P. jirovecii* in a respiratory sample. Medically relevant information such as demographical characteristics, clinical presentation, microbiological co-infections, and treatments were collected. The objectives were to describe the characteristics of these children with *P. jirovecii* colonization/infection to determine the key underlying diseases and risk factors, and to identify viral respiratory pathogens associated. The PCR was positive for *P. jirovecii* in 32 children. Cardiopulmonary pathologies (21.9%) were the most common underlying disease in them, followed by severe combined immunodeficiency (SCID) (18.8%), hyaline membrane disease (15.6%), asthma (9.4%) and acute leukaemia (6.3%). All SCID children were diagnosed with *pneumocystis* pneumonia. Co-infection with Pj/Rhinovirus (34.4%) was not significant. Overall mortality was 18.8%. Paediatric *pneumocystis* is not restricted to patients with HIV or SCID and should be considered in pneumonia in children under 3 years old.

## 1. Introduction

The opportunistic fungus *Pneumocystis jirovecii* can cause Pneumocystis pneumonia (PCP) in immunosuppressed patients, with an annual incidence of 19.4 cases per million inhabitants according to recent European data [1]. Acquisition of *P. jirovecii* infection early in life has been confirmed by PCR and serologic studies; more than 80% of children between the ages of two and four years-old have been exposed to *P. jirovecii* [2,3,4]. In children, the main predisposing factors for developing PCP include acquired immune deficiency syndrome (AIDS), immunosuppressive therapy, severe combined immunodeficiency (SCID) and preterm birth [5,6]. However, with the rising number of immunocompromising conditions, *P. jirovecii* is increasingly recognized as an opportunistic pathogen in children. Some underlying diseases, such as congenital disorders impacting the respiratory tree, should be also considered. However, the significance of *P. jirovecii* harboured in the respiratory tracts of non-severely immunosuppressed children is more difficult to interpret. In fact, some studies showed that primary infection in non-severely immunosuppressed children can be symptomatic, with a self-limiting upper respiratory tract infection in infants [2,5]. The border between colonization and infection in these patients therefore remains unclear. Interestingly, Nevez et al. reported a high rate (60%) of viral infection among PCP infants, with some defined elements regarding virus panel tests, severity and outcomes in children. Moreover, in adult patients, earlier studies identified a significant association between human rhinovirus (hRV) and *P. jirovecii* [7]. This French retrospective cohort study aimed to describe features of the *P. jirovecii*-positive children under three years old by insisting on the description of the underlying diseases, the clinical data, the evolution and the description of the viral respiratory pathogens associated with the detection of *P. jiroveccii*.

## 2. Materials and Methods

### 2.1. Study Location

The site for this retrospective study was La Timone university hospital, a 276-bed paediatric tertiary care institution located in Marseille, France. Patients were treated in the 24 specialized paediatric departments, including haematology, oncology, surgery, intensive care units and multidisciplinary paediatric units.

### 2.2. Primary Objective

The primary objective was to describe the characteristics of children under 3 years old associated with the detection of *P. jirovecii* (Pj) by PCR.

Consequently, all children under 3 years old treated in the same medical units from April 2014 to August 2020 and for whom a *P. jirovecii* evaluation had been undertaken were enrolled in the study. Two groups were created: (1) the Pj positive group included all children with positive *P. jirovecii* detection; (2) the Pj negative group included all children with negative *P. jirovecii* detection. A positive case was defined as at least one positive *P. jirovecii* PCR result from a respiratory tract sample, including throat swabs, nasopharyngeal swabs, sputum, bronchial aspirate, and broncho-alveolar fluid (BALF). Given the retrospective collection of data, direct examination of respiratory samples was not available. All DNA samples came from a syndromic diagnosis based only on molecular diagnosis within the framework of investigation of pulmonary symptoms or fever. Discriminating between PCP and pulmonary *P. jirovecii* colonization is not possible based on PCR results.

Medically relevant information was compiled: general demographic information, underlying conditions, clinical and radiological presentation, microbiological co-infections, HIV status and medical treatments. Clinical presentations were divided into upper (rhinitis, cough, asthma, and nasopharyngitis) and lower respiratory tract infections (bronchiolitis, atelectasis, dyspnoea, oxygen desaturation, acute respiratory distress syndrome, pulmonary imaging abnormality). The role of *P. jirovecii* colonization/infection in clinical progression was evaluated by the number of days of hospitalization, prescription of trimethoprim/sulfamethoxazole (TMP-SMX), evolution of oxygen saturation, and overall *Pneumocystis*-related mortality. The final medical report following concomitant care for all children with a *P. jirovecii*-positive sample helped to classify patients as *P. jirovecii* colonization status or infection status, according to improvement after TMP-SMX medication, radiological data and physician conclusion (Appendix A), within the limits of current knowledge and available tools.

### 2.3. Secondary Objective

The secondary objective was to identify viral respiratory pathogens associated with the detection of *P. jirovecii* by PCR.

Using a syndromic approach facilitating the most exhaustive detection of respiratory virus, data were collected between January 2017 and December 2019, within a maximum interval of seven days from *P. jirovecii* sampling. Two groups were created. The positive group included children with a *P. jirovecii*-positive respiratory sample. The negative control group included children with a lower or upper respiratory syndrome not associated with *P. jirovecii* detection. All children were treated in the same medical units and presented similar underlying diseases.

### 2.4. Laboratory Processing

#### 2.4.1. Pj Detection in Clinical Samples

*P. jirovecii* DNA was extracted using the NucliSens^TM^ EasyMAG^TM^ technology (BioMérieux, Marcy-l’Étoile, France). PCR amplification targeting the mitochondrial large subunit ribosomal RNA gene using the forward primer pAZ102-H (5′-GTGTACGTTGCAAAGTACTC-3′), the reverse primer pAZ102-E (5′-GATGGCTGTTTCCAAGCCCA-3′) and an in-house probe (6FAM-TCTGGGCTGTTTCCCTTTCGACT) [8] was carried out using the LightCycler^®^ 480 Probes Master kit (Roche Diagnostics, Meylan, France) according to the manufacturer’s recommendations. A negative control (water), positive control (DNA from plasmids containing the target sequence), and extraction controls (albumin gene) were included in each run.

#### 2.4.2. Viral DNA and RNA Extraction

DNA and RNA were extracted from the respiratory samples using the EZ1 Advanced XL (Qiagen, Hilden, Germany) with the Virus Mini Kit v2.0 (Qiagen, Hilden, Germany), according to the manufacturer’s recommendations.

#### 2.4.3. Detection of Respiratory Virus in Clinical Samples

PCR detection of viruses was performed (a) from April to October using the FTD Respiratory Pathogens 21-plus kit (Fast Track Diagnostics, Luxembourg) for human rhinovirus (hRV), human influenza virus A/B, human parainfluenza virus (hPiV) type 1, 2, 3 and 4, human coronavirus (hCoV) NL63, 229E, OC43, HKU1, human respiratory syncytial virus A/B (hRSV), human adenovirus (hAdV) and human enterovirus from all groups (hEV), and (b) from November to March using single-plex assays for human rhinovirus (hRV) (targeting 5’ UTR (untranslated region), in-house), human respiratory syncytial virus A/B (hRSV), human adenovirus (hAdV), human enterovirus from all groups (hEV), human metapneumovirus (hMPV), and human influenza virus A/B. One-step duplex quantitative qPCR amplifications of hCoV/hPiV-R Gene Kit (BioMérieux, Marcy l’Etoile, France) were used to detect hCoV and hPiV, according to the manufacturer’s recommendations [9,10,11,12,13]. A negative control (water), positive control (DNA from plasmids containing the target sequence), and extraction controls were included in each run.

### 2.5. Statistical Analysis

Statistical analyses were performed using SAS 9.4 statistical software. Qualitative variables were described using frequencies, and their association with *Pneumocystis jirovecii* was tested using the Fisher exact test. A two-sided *p*-value less than 0.05 was considered statistically significant.

### 2.6. Ethical Considerations

This non-interventional study did not require ethical approval or informed consent, according to French laws and regulations (CSP Art L1121e1.1). The study was approved by the Assistance Publique des Hôpitaux de Marseille (APHM) Institutional Ethics Committee on 29 May 2019, number 2019-73.

## 3. Results

### 3.1. Primary Objective: Characteristics of Pj Positive Children

Between April 2014 and August 2020, 279 children under 3 years old had a *P. jirovecii* assay in respiratory samples, in whom 32 were positive (five nasal or pharyngeal swabs, two sputum, seven naso-pharyngeal aspirates, six bronchial aspirates, eight BAL, one lung biopsy and three unknown) (Table 1). No seasonality was observed. The average age was 7 months, and the sex ratio (M/F) was 0.88. The majority (71.9%) of infants were born at term. Among the underlying diseases, cardiopulmonary pathologies were predominant (21.9%), followed by SCID (18.8%), hyaline membrane diseases (15.6%), other congenital abnormalities (15.6%), mild asthma (9.4%) and acute leukaemia (6.3%). It is noteworthy that only one child had no underlying disease. No significant difference was observed between the Pj positive and the Pj negative groups concerning the underlying diseases, with the exception of the absence of underlying disease, found mainly in the Pj negative group (*p* = 0.0143). Seven children were receiving corticosteroid treatment (one oral, one intravenous and five respiratory route) at the time of Pj positive detection. Two children had never been discharged from the hospital between birth and the first Pj positive sample. Regarding pulmonary symptoms, 78.1% presented lower respiratory symptoms, 31.2% upper respiratory symptoms and 6.2% no respiratory symptoms. Oxygen saturation was measured in 21 patients over the three days surrounding the diagnosis, ranging from 47% to 100% of SpO2 levels (average value 92%). Serum LDH was measured in eight patients (mean value: 449 IU/L; range 221–971), of whom six had a higher LDH level (cut-off: <248 IU/L). Ten patients were hospitalized in the intensive care unit.

According to medical reports analysed for each of the 32 patients, PCP was diagnosed in 18 patients (56.3%), Pj colonization in 13 patients (40.6%) and one patient (3.1%) was unclassified because death occurred soon after hospital admission (Appendix A). Curative treatment with TMP-SMX was initiated for all PCP patients. These children were mainly affected by congenital diseases, mostly SCID, and two of them were born preterm (both at 34 WG). Interestingly, one child had only asthma as an underlying disease. A similar profile of underlying disease was observed between the PCP and Pj colonization group, except that all children who had SCID were diagnosed with PCP.

Overall mortality was 18.8% (six patients). One death was directly imputable to PCP. The others were related to their unfortunate underlying disease, known to have a short life expectancy.

### 3.2. Secondary Objective: Identification of Viral Pathogens Associated with Pj in Respiratory Samples

Between January 2017 and December 2019, viral pathogen detection in respiratory samples could be collected for Pj positive patients (*n* = 13) and for negative control patients (*n* = 31) (Table 2). The mean age and the sex ratio were comparable between the two groups. A univariate analysis was performed as a means of identifying co-infection viruses associated with Pj (Table 2). The percentage of children with a respiratory co-infection virus plus Pj was 53.9%, and the percentage of children with virus infection without Pj was 45.2% (*p* = 0.7438). hRV was the most frequent agent isolated. However, neither hRV nor any other virus was significantly associated with Pj positive detection (Table 2). Among the Pj positive group, seven were classified as PCP, five as Pj colonization and one remained unclassified as explained above. It did not appear that viral agents were associated with the clinical form of PCP.

## 4. Discussion

In this retrospective study, we have focused on the characteristics of children under 3 years old in whom *Pneumocystis jirovecii* was detected by PCR in upper and lower respiratory samples. Direct examination of respiratory samples was not possible and not performed retrospectively. Few publications are specifically interested in this topic, and information concerning *P. jirovecii* is often obscured by the other pathogens responsible for pneumonia. To our knowledge, this is one of the largest studies specifically focusing on a cohort of children under the age of three with an exhaustive description of underlying diseases. All children taken care of in the paediatric units were included. PCR tests for *P. jirovecii* from respiratory samples were positive in 32/279 (11.4%) children under 3 years old. In the literature, the available data regarding *Pneumocystis* molecular detection in the infant population have reported positive rates of between 16% and 32% [2,5,14,15,16]. However, it is difficult to compare these results because the medical and demographic data are scattered, especially age distribution and underlying diseases. Nevez et al. found an 18.2% of positive rate with a cohort most comparable to ours [5]. Historically, preterm birth is the primary risk factor described [17]. In our study, 71.8% of children were born at term. Nevez et al. also showed that 71.4% of children presenting a *P. jirovecii*-positive respiratory sample were born at term, and the authors did not identify prematurity as a risk factor associated with *P. jirovecii* [5]. Furthermore, among the cases reports of PCP in children under 3 years old, the most frequently found underlying disease is SCID [18,19,20,21,22]. Indeed, we observed that Pj-positive status was not always associated with severe immunosuppression. Nevez et al. described Pj in a non-immunosuppressed infant population with underlying diseases comprising congenital diseases, hyaline membrane disease/bronchopulmonary dysplasia and cystic fibrosis [5]. In the present study, the underlying diseases were predominantly cardiopulmonary pathologies (21.9%) followed by SCID (18.8%), hyaline membrane disease (15.6%) and other congenital abnormalities (15.6%). The remaining question is the interpretation of molecular Pj detection in a child under 3 years old with underlying diseases other than immunosuppressive diseases. Pj colonization status may correspond to a primary infection, as described in 20% of immunocompetent children between two and four years old, some with symptomatic forms [2,16]. Notably, data from different studies suggest a potential role of *P. jirovecii* in the progression of chronic pulmonary diseases [23,24,25]. In our cohort, 16 children in the Pj positive group had cardiopulmonary pathologies and chronic pulmonary pathologies such as asthma, hyaline membrane disease and bronchiolitis. One child, classified as Pj colonization, developed an acute respiratory distress syndrome secondary to Pj positive detection. This suggests that this pathogen must be considered even in the event of colonization. Non/low-immunocompromised children are susceptible to developing PCP: we found that 56.3% of Pj-positive patients were diagnosed with PCP according to physician medical reports. In this group, the underlying diseases were SCID and acute leukaemia but also various congenital diseases. Interestingly, one PCP child had asthma. This child presented with persistent wheezing for 3 months, with two episodes of exacerbation despite appropriate treatment. He had a good clinical course after treatment with TMP-SMX, corticosteroid therapy and long-term asthma control medications. This is consistent with recent studies suggesting that Pj can plays a role in both the pathophysiology and the exacerbation of asthma [24,25]. Importantly, all patients with SCID were considered and treated as PCP, which confirms that the presence of a *P. jirovecii*-positive specimen in a patient with SCID should be considered as an infection in progress. Finally, no underlying conditions were particularly associated with the positive detection of Pj in a respiratory sample, in comparison with the Pj negative group.

The overall mortality rate was 18.8% and only one death was attributable to PCP. Other studies show a similar mortality rate, ranging from 11.5% to 20% in children under 5 years-old treated for severe PCP [18,26].

The paediatric population is known to be vulnerable to viral and bacterial infections, due to the immaturity of the immune system [27]. The respiratory pathogen most frequently concomitantly isolated with *P. jirovecii* was hRV (6/13 patients). However, the association between hRV and Pj was not significant, and none of the other viral respiratory pathogens were significantly associated with *P. jirovecii.* These results are consistent with other studies. Larsen et al., in a blinded retrospective study on children under 24 months of age with acute respiratory tract infection, found no significant association between hRSV and Pj [16]. Likewise, Lanaspa et al. found a similar distribution of respiratory viruses in PCP and negative-PCP children under five years-old hospitalized with severe pneumonia [28].

High-dose TMP-SMX is recommended as first-line treatment for PCP [29]. In our children classified as Pj colonization status, only one received curative treatment. This suggests that a positive PCR result is not always considered, whatever the immune status or the term at birth. As However, Pj colonization may be a risk factor for developing respiratory distress syndrome, mainly in preterm births [30]. In fact, one asthmatic child in the Pj colonization group developed an exacerbation of his asthma concomitantly with positive *P. jirovecii* detection. The systematic treatment of patients with a positive PCR sample could perhaps be considered. However, physicians are cautious because of side effects [31]. It is necessary to assess the risk/benefit ratio when considering a *P. jirovecii*-positive result.

## 5. Conclusions

The take home message from this study is that detection of *Pneumocystis jirovecii* in respiratory samples in children under 3 years old should be considered as a potentially infectious process. Some children worsened following Pj diagnosis, with progression to acute respiratory symptoms or death. Moreover, Pj may be the only respiratory pathogen found during the management of respiratory infections. In the era of syndromic diagnosis, numerous respiratory panels are available, allowing the simultaneous detection of at least five pathogens, including viruses and bacteria, except for *P. jirovecii* [32]. The results presented in this paper highlight the need to screen for *P. jirovecii* in cases of non-specific respiratory symptoms in children under 3 years old. A systematic evaluation for *Pneumocystis jirovecii* would help to better understand its involvement in respiratory symptoms in these very young patients.

## Figures and Tables

**Table 1 jof-07-00546-t001:** Description of the child population (2014–2020).

		Pj Positive Group			Pj Negative Group			*p*
		*n*	%	95% CI	*n*	%	95% CI	
Child characteristics								
Number of PCR Pj tests		32			247			
Sex ratio (M/F)		0.88			1.12			
Mean age (months)		7			11.5			
Birth term	>37 WG	23	71.9		178	72.1		
	33–37 WG	6	18.8		46	18.6		
	28–32 WG	2	6.2		13	5.3		
	<28 WG	1	3.1		10	4.0		
Blood-related parents		7	21.9	0.0928–0.3997	15	6.1	0.0344–0.0982	0.0066
HIV-positive		0	0		0			
Underlying diseases								
Severe combined immune deficiency		6	18.8	0.0721–0.3644	27	10.9	0.0733–0.1550	0.0871
Cardiopulmonary pathologies		7	21.9	0.0928–0.3997	48	19.4	0.1216–0.2085	0.8135
	Cardiopathies	5	15.6		8	3.2		
	Inter-atrial and/or interventricular communication	1	3.1		16	6.5		
	Tetralogy of Fallot	1	3.1		7	2.8		
	Transposition of the great vessels	0	0		5	2.0		
	Other vessel abnormalities	0	0		2	0.8		
	Bicuspid valve	0	0		2	0.8		
	Ebstein’s anomaly	0	0		1	0.4		
	Marfan’s syndrome	0	0		1	0.4		
	CHARGE syndrome	0	0		1	0.4		
	Neuhauser syndrome	0	0		1	0.4		
	ARCAPA syndrome	0	0		1	0.4		
	Trisomy 4	0	0		1	0.4		
	Pulmonary arterial hypertension	0	0		1	0.4		
	Kawasaki disease	0	0		1	0.4		
Metabolic abnormalities		1	3.1	0.0008–0.1622	5	2.0	0.0066–0.0466	0.5233
	NFU1 deficiency	1	3.1		0	0		
	Farber disease	0	0		2	0.8		
	Krabbe disease	0	0		1	0.4		
	Mevalonate kinase deficiency	0	0		1	0.4		
	Metabolic syndrome	0	0		1	0.4		
Acute leukaemia		2	6.3	0.0077–0.2081	18	7.3	0.0438–0.1127	1.0000
	Acute myeloid leukaemia	2	6.3		6	2.4		
	Acute lymphoid leukaemia	0	0		11	4.4		
	Juvenile myelomonocytic leukaemia	0	0		1	0.4		
Other Hematologic malignancies		0	0	0.0000–0.1089	8	3.2	0.0141–0.0628	1.0000
Solid cancer|Lymphoma		1	3.1	0.0008–0.1622	20	8.1	0.0502–0.1223	0.4854
	Fibromatosis	1	3.1		1	0.4		
	Neuroblastoma	0	0		6	2.4		
	Nephroblastoma	0	0		2	0.8		
	Lymphoma	0	0		2	0.8		
	Hepatoblastoma	0	0		1	0.4		
	Retinoblastoma	0	0		1	0.4		
	Rhabdomyosarcoma	0	0		1	0.4		
	Astrocytoma	0	0		1	0.4		
	Other solid tumour	0	0		5	2.0		
Autoimmune disease		0	0	0.0000–0.1089	5	2.0	0.0066–0.0466	1.0000
Cystic fibrosis|Hyaline membrane disease|Alveolar proteinosis		5	15.6	0.0528–0.3279	14	5.7	0.0313–0.0933	0.1383
	Hyaline membrane disease	5	15.6		7	2.8		
	Cystic fibrosis	0	0		5			
	Alveolar proteinosis	0	0		2	0.8		
Asthma|Respiratory pathology isolated		4	12.5	0.0351–0.2899	30	12.1	0.0835–0.1688	1.0000
	Asthma	3	9.4		21	8.5		
	Bronchopulmonary dysplasia	0	0		7	2.8		
	Pulmonary valve stenosis	0	0		1	0.4		
	Bronchiectasis	0	0		1	0.4		
	Post infectious obstructive bronchiolitis	1	3.1		0	0		
Other congenital abnormalities		5	15.6	0.0528–0.3279	22	8.9	0.0567–0.1317	0.5176
	Biliary atresia	1	3.1		6	2.4		
	Polymalformative syndrome	0	0		6	2.4		
	Diaphragmatic hernia	0	0		2	0.8		
	Infantile nephrotic syndrome	1	3.1		0	0		
	Pierre Robin syndrome	1	3.1		0	0		
	HTZ mutation	1	3.1		0	0		
	Familial dyserythropoiesis	1	3.1		0	0		
	Pyelo-caliceal dilatation	0	0		1	0.4		
	Encephalopathy	0	0		1	0.4		
	Severe laryngomalacia	0	0		1	0.4		
	Giant omphalocele	0	0		1	0.4		
	Beckwith-Wiedemann-Syndrome	0	0		1	0.4		
	Crouzon syndrome	0	0		1	0.4		
	End-stage renal disease	0	0		1	0.4		
	Situs inversus	0	0		1	0.4		
Solid organ transplant		0	0	0.0000–0.1089	2	0.8	0.0010–0.0289	1.0000
No underlying disease (except for prematurity)		1	3.1	0.0008–0.1622	52	21.1	0.1614–0.2667	0.0143
Clinical manifestations and management								
Clinical manifestations								
Upper respiratory tract infection		10	31.2	0.1612–0.5001	85	34.4	0.2851–0.4070	0.8436
Lower respiratory tract infection		25	78.1	0.6003–0.9072	106	42.9	0.3666–0.4934	0.0002
No respiratory symptoms		2	6.2	0.0077–0.2081	61	24.7	0.1945–0.3056	0.0224
Fever		10	31.3	0.1612–0.5001	91	36.8	0.3081–0.4319	0.5646
Clinical management								
TMP-SMX curative treatment		18	56.2		NA	NA		
Radiological evidence of infectious process		19	59.4	0.4065–0.7630	41	16.6	0.1218–0.2184	1.0000
Mortality								
Overall mortality		6	18.8	0.0721–0.3644	22	8.9	0.0567–0.1317	0.1102
90-day mortality		5	15.6		NA	NA		
30-day mortality		4	12.2		NA	NA		

Pj: *Pneumocystis jirovecii*; CI: Confidence interval; M: Male; F: Female; WG: Week of gestation; HIV: Human immunodeficiency virus; TMP-SMX: trimethoprim/sulfamethoxazole; NA: Not applicable.

**Table 2 jof-07-00546-t002:** Description of viral coinfection in the Pj positive group and the negative control group (2017–2019).

		Pj Positive Group			Negative Control Group			*p*
		*n*	%	95% CI	*n*	%	95% CI	
Child characteristics								
Number of children		13			31			
Sex ratio (M/F)		0.86			1.21			
Mean age (months)		7.3			10.39			
Birth term	>37 WG	10	76.9		23	74.2		
	32–37 WG	3	23.1		3	9.7		
	28–32 WG	0	0		2	6.5		
	<28 WG	0	0		3	9.7		
HIV-positive		0	0		0	0		
Viral respiratory pathogens								
Number of children with viral respiratory pathogens		7	53.9	0.2513–0.8078	14	45.2	0.2732–0.6397	0.7438
	Rhinovirus	6	85.71	0.4213–0.9964	10	71.4	0.4190–0.9161	0.4964
	Adenovirus	1	14.3	0.0036–0.5787	5	35.7	0.1276–0.6486	0.6523
	Parainfluenza Virus	0	0		2 ^a^	14.3	0.0178–0.4281	1.0000
	Coronavirus	0	0		1 ^b^	7.1	0.0018–0.3387	1.0000
	Metapneumovirus	0	0		1	7.1	0.0018–0.3387	1.0000
	Enterovirus	2	28.6	0.0367–0.7096	0	0		0.0825
	Respiratory Syncytial Virus	2	28.6	0.0367–0.7096	0	0		0.0825
Number of children without viral respiratory pathogens		6	46.2		17	54.8		

Pj: *Pneumocystis jirovecii*; CI: Confidence interval; M: Male; F: Female; WG: Week of gestation; HIV: Human immunodeficiency virus. ^a^ Parainfluenzae type 1 (*n* = 1) and type 3 (*n* = 1); ^b^ Coronavirus E229.

## Data Availability

All data are available within the article and Appendix A.

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
