# Peer review of "Detection of Pneumocystis jirovecii in Hospitalized Children Less Than 3 Years of Age"

_jof, 2021, doi:10.3390/jof7070546_

Round 1

Reviewer 1 Report

I read with interest the manuscript by Menu and colleagues entitled “Pneumocystis jirovecii in children under 3 years old: characteristics, co-infections, and review of the literature.”

The main strength of this article is to provide additional data on Pneumocystis epidemiology among children. The main weakness is that the criteria used to distinguish between PCP and colonization are really confusing.

Major comments

1) Distinction between colonization and PCP is not clear; this issue needs to be thoroughly reviewed throughout the manuscript.

2) The main articles regarding Pneumocystis detection using PCR assays in children or infants have described positivity rates of 32 % (doi: 10.1086/319340), 25.3 % (doi: 10.1111/j.1550-7408.2001.tb00479.x.), 24.5% (doi: 10.1111/j.1550-7408.2003.tb00678.x) 16 % (doi: 10.3201/eid1301.060315) and 18.2 % (doi: 10.1093/mmy/myz040). In the present study, this rate is 11.4%. As the authors refer to a review of the literature in the article title, they should -at least- consider these differences.

Other comments and queries

The concepts of colonization and primary infection should be addressed in the introduction.

L 30. Please refer to the acronym “PCP” as “Pneumocystis pneumonia” and not as “severe pneumonia”.

L 33. Please, add the first references to serological studies which have established that primary Pneumocystis infection occurs early in life: PMID 328785 and PMID 400818.

L 49. Please replace “patient” by “children”.

L 67. Please explain why direct examination of respiratory samples, especially BALF specimens, was not performed.

L 68. The authors state that discriminating between PCP and colonization is not possible based on PCR results. However, Ct values could be added and discussed.

L 77. Italicize Pneumocystis

L 77-80. Once again, the authors should be more precise regarding criteria used to colonization/PCP distinction. What are the radiological data?

L 87. What was the reason for the 7-day interval?

L 99. Wakefield et al. have described the 2 primers; please add a reference for the probe.

L 102. What type of extraction control was used?

L 128. Italicize Pneumocystis jirovecii

L 131. It is right that this non-interventional study did not required informed consent. However, according to the French Data Protection Authority (CNIL), the study must be registered by an institutional review board.

L 199. Larsen et al. (doi: 10.3201/eid1301.060315) have included more than 400 infants and their study was the largest one.

L 214-215. The phrase « the remaining question …well described » is confusing. Please rephrase.

L 234. Replace « Pj pneumonia » by PCP

L 246-247. Clinical improvement in the absence of anti-Pneumocystis treatment is one of the criteria for a diagnosis of colonization. A child who has received cotrimoxazole cannot be considered as colonized by Pneumocystis.

Author Response

Answer to the reviewers:

Review 1

I read with interest the manuscript by Menu and colleagues entitled “Pneumocystis jirovecii in children under 3 years old: characteristics, co-infections, and review of the literature.”

The main strength of this article is to provide additional data on Pneumocystis epidemiology among children. The main weakness is that the criteria used to distinguish between PCP and colonization are really confusing.

Response: We thank the reviewer for his/her nice comment. We agree with the comment about distinction between PCP and colonization. We are working to clarify and nuanced this particular point.

Major comments

1) Distinction between colonization and PCP is not clear; this issue needs to be thoroughly reviewed throughout the manuscript.

Response 1: We have clarify the distinction between PCP and colonization in the Materials and Methods section and we have nuanced it throughout the manuscript. At present, the distinction between colonization and infection remains difficult and it is mainly thanks to a positive direct examination or to hard clinical arguments that the PCP diagnosis is retained. Unfortunately, it is a retrospective study, and the clinical samples are essentially upper respiratory specimens for which direct examination is not informative because of an obvious lack of sensitivity.

All the patients' medical records were reviewing one by one. As mentioned in the Materials and Methods section we took into account "to classify patients as P. jirovecii colonization status or infection status, according to improvement after TMP-SMX medication, radiological data and physician conclusion (Supplementary Table 1)". All information is available in the supplementary table 1 data. However, because we are aware that this classification can be discussed, we have specified " … , within the limits of current knowledge and available tools.”. (Line 95-99)

2) The main articles regarding Pneumocystis detection using PCR assays in children or infants have described positivity rates of 32 % (doi: 10.1086/319340), 25.3 % (doi: 10.1111/j.1550-7408.2001.tb00479.x.), 24.5% (doi: 10.1111/j.1550-7408.2003.tb00678.x) 16 % (doi: 10.3201/eid1301.060315) and 18.2 % (doi: 10.1093/mmy/myz040). In the present study, this rate is 11.4%. As the authors refer to a review of the literature in the article title, they should -at least- consider these differences.

Response 2: We agree with the reviewer and add in the discussion section (Line 223-227): “ In the literature, the available data regarding Pneumocystis molecular detection in infant population have reported positive rates between 16 to 32% (PMID: 11247708; Nevez 2001; Nevez 2020; Larsen). But it is difficult to compare together these data because the medical and demographic data are scattered especially age distribution and underlying diseases.  Nevez et al. find 18.2% of positive rate with a cohort most comparable to ours”.

And as suggested by the reviewer 2 (“the title does not represent the manuscript. "review of the literature" needs to be deleted. For example:  Detection of Pneumocystis jirovecii in hospitalized children less than 3 years of age. ), we change the title as “Detection of Pneumocystis jirovecii in hospitalized children less than 3 years of age”.

 Other comments and queries

The concepts of colonization and primary infection should be addressed in the introduction.

Response: we have modified your introduction for this purpose

L 30. Please refer to the acronym “PCP” as “Pneumocystis pneumonia” and not as “severe pneumonia”.

Response: We agree with the reviewer and refer as “Pneumocystis pneumonia (PCP)” (Line 32)

L 33. Please, add the first references to serological studies which have established that primary Pneumocystis infection occurs early in life: PMID 328785 and PMID 400818.

Response: We agree with the reviewer and add the both references (Line 36)

L 49. Please replace “patient” by “children”.

Response: We agree with the reviewer and replace “patient” by “children” (Line 49)

L 67. Please explain why direct examination of respiratory samples, especially BALF specimens, was not performed.

Response: We have more explain this point as “Given the retrospective collection of data, direct examination of respiratory samples was not available. All DNA samples came from a syndromic diagnosis based only in molecular diagnosis within the framework of investigation of pulmonary symptoms or fever.” (Line 80).

L 68. The authors state that discriminating between PCP and colonization is not possible based on PCR results. However, Ct values could be added and discussed.

Response: We agree that some publications tend to show that the Ct value can point to an infection and colonization. But essentially when it comes to BAL type samples.

In our study, as we are working on samples from the upper (throat swabs, nasopharyngeal swabs, sputum, bronchial aspirate) and lower (broncho-alveolar fluid) respiratory tract, it is not possible to compare Ct values between them. This is particularly true for upper respiratory tract specimens because in this case the Ct may be late despite an infection status.

L 77. Italicize Pneumocystis

Response: We italicize Pneumocystis (Line 94)

L 77-80. Once again, the authors should be more precise regarding criteria used to colonization/PCP distinction. What are the radiological data?

Response: More details are provided on the radiological data in the Supplementary Table 1 (referred line 98). In this supplementary table, we change the title “Pulmonary imagery” by “Radiological data” in a concern of clearness and uniformity of the manuscript.

L 87. What was the reason for the 7-day interval?

Response: This 7-day interval was chosen in order to try to be as contemporary as possible to the detection of Pneumocystis by PCR. Indeed, we aimed to identify co-infections from the same episode and not super-infections.

L 99. Wakefield et al. have described the 2 primers; please add a reference for the probe.

Response: We forgot to mention that the probe is design in-house. This information has been added in the Materials and Methods section (Line 117).

L 102. What type of extraction control was used?

Response: We forgot to mention that the extraction control is the albumin gene. We have added this information in the Materials and Methods section (Line 120).

L 128. Italicize Pneumocystis jirovecii

Response: We italicize Pneumocystis jirovecii (Line 146)

L 131. It is right that this non-interventional study did not required informed consent. However, according to the French Data Protection Authority (CNIL), the study must be registered by an institutional review board.

Response: We added in the 2.6. Ethical consideration section “The study was approved by the AP-HM Institutional Ethics Committee on May 29, 2019, number 2019-73.”

L 199. Larsen et al. (doi: 10.3201/eid1301.060315) have included more than 400 infants and their study was the largest one.

Response: In the Larsen et al. study, the age of the children was 49 to 265 days. This publication is cited in our references. In the literature about Pneumocystis detection in infant it is rare to find cohorts of children aged 9 months to 3 years. However, in order to considerate our remarks, we specify “To our knowledge, this is one of the largest study specifically focusing on a cohort of children under the age of three.” (Line 219).

L 214-215. The phrase « the remaining question …well described » is confusing. Please rephrase.

Response: In order to clarify this sentence we have reformulated it as follows “The remaining question is the interpretation of Pj detection in a child under 3-years-old with underlying diseases other than immunosuppressive diseases.” (line 243)

L 234. Replace « Pj pneumonia » by PCP

Response: We replace “Pj pneumonia” by “PCP” (Line 268)

L 246-247. Clinical improvement in the absence of anti-Pneumocystis treatment is one of the criteria for a diagnosis of colonization. A child who has received cotrimoxazole cannot be considered as colonized by Pneumocystis.

Response: We agree with the reviewer, this is a mistake on our part. This is patient 31, we have reclassified him as an infection. All data were verify and recalculated.

Reviewer 2 Report

This retrospective study contributes to understand whether Pneumocystis detection has a role in disease in children younger than 3 years of age.

The title does not represent the manuscript. "review of the literature" needs to be deleted. For example:  Detection of Pneumocystis jirovecii in hospitalized children less than 3 years of age. 

Minor comments:

Line 33 reads: ...have been confirmed by serologic...

Should read: ...have been confirmed by PCR and serologic... (The reference is fine)

Section 2.2

The terms colonization/infection may be confusing as used. Suggest to delete this term and instead use: The primary objective was to describe the characteristics of children under 3 years old associated with detection of P. jirovecii by nested PCR. 

"detection" is how the authors refer in further sections of the manuscript and should be applied also, to section 2.3. Would recommend to omit the distinction between colonization and infection that for this methodology is confusing and agree that definition of these terms does not contribute in the context of this study. 

Lines 160 includes comments on a child with asthma that should be moved to the discussion section. Please add reference by Eddens. 

The discussion section can be improved with a . These are suggestions of articles to be considered:  

Walzer PD  Natl Cancer Inst Monogr. 1976 Oct;43:55-63; Morrow BM  BMC Research Notes 2014; Vargas SL J Infect Dis 2015; Taylor Eddens Ann Allergy Asthma Immunol 2021; 

Author Response

Answer to the reviewers:

Review 2

This retrospective study contributes to understand whether Pneumocystis detection has a role in disease in children younger than 3 years of age.

Response: We thanks the reviewer 2 for his/her nice comment.

The title does not represent the manuscript. "review of the literature" needs to be deleted. For example:  Detection of Pneumocystis jirovecii in hospitalized children less than 3 years of age.

Response: We change the title as “Detection of Pneumocystis jirovecii in hospitalized children less than 3 years of age”

Minor comments:

Line 33 reads: ...have been confirmed by serologic...

Should read: ...have been confirmed by PCR and serologic... (The reference is fine)

Response: We add “… by PCR and …” (Line 35)

Section 2.2

The terms colonization/infection may be confusing as used. Suggest to delete this term and instead use: The primary objective was to describe the characteristics of children under 3 years old associated with detection of P. jirovecii by nested PCR.

R: We have considerate this suggestion and modified as follow:  “The primary objective was to describe the characteristics of children under 3 years old associated with detection of P. jirovecii by PCR.” (Line 73). We have not added “nested” because we have used a one-step a real time PCR.

"detection" is how the authors refer in further sections of the manuscript and should be applied also, to section 2.3. Would recommend to omit the distinction between colonization and infection that for this methodology is confusing and agree that definition of these terms does not contribute in the context of this study.

Response: We agree with the reviewer and modified as follow:  “The secondary objective was to identify viral respiratory pathogens associated with the detection of P. jirovecii by PCR.” (Line 101)

Lines 160 includes comments on a child with asthma that should be moved to the discussion section. Please add reference by Eddens.

Response: We agree with the reviewer. We read with interest the articles concerning the relationship between Pneumocystis jiroveccii and asthma. As suggested, we have moved the comments on a child with asthma that to the discussion section and added the reference “Taylor Eddens Ann Allergy Asthma Immunol 2021”.

As suggested by the reviewer we add “This child presented with persistent wheezing for 3 months, with two episodes of exacerbation despite appropriate treatment. He had a good clinical course after treatment with TMP-SMX, corticosteroid therapy and long-term asthma control medications. This is consistent with recent studies suggesting that P. jirovecii can plays a role in both the pathophysiology exacerbation of asthma [24, 25]” in the discussion section. (Line 255)

The discussion section can be improved with a . These are suggestions of articles to be considered: 

Walzer PD  Natl Cancer Inst Monogr. 1976 Oct;43:55-63; Morrow BM  BMC Research Notes 2014; Vargas SL J Infect Dis 2015; Taylor Eddens Ann Allergy Asthma Immunol 2021;

Response: Thank you for this suggestions. We have found another recent reference on the relationship between Pneumocystis and asthma and have added it to our discussion. We take the liberty of not citing Morrow et al. because the cohort described is not comparable to ours (HIV patients) and Walzer et al. because the publication presents out of dated diagnostic tools.

Round 2

Reviewer 1 Report

I thank the authors for the revised version.

In my opinion, the manuscript has been significantly improved .